# A Case of Diffuse Thyroid Hematoma after Ultrasound-Guided Fine Needle Aspiration

**DOI:** 10.3390/medicina59040690

**Published:** 2023-03-31

**Authors:** Jong Gyu Lee, Young-Soo Chang, Bo Young Kim

**Affiliations:** Departments of Otorhinolaryngology-Head and Neck Surgery, Inje University Sanggye Paik Hospital, Seoul 01757, Republic of Korea; ljk10216@naver.com (J.G.L.); s5636@paik.ac.kr (Y.-S.C.)

**Keywords:** ultrasound-guided fine needle aspiration, thyroid hematoma, edoxaban

## Abstract

Ultrasound-guided fine needle aspiration is an easy, safe, and efficient method of diagnosing thyroid diseases. Recent guidelines and studies have demonstrated that this test has a low incidence of complications; thus, most guidelines do not provide recommendations for post-exam care. However, the risk of serious and fatal bleeding in selected patients with bleeding tendency exists. Although screening tests for coagulation are not always necessary, a thorough assessment of past medical history needs to be made to identify disorders affecting coagulation function and bleeding risk factors, such as the use of antithrombotic drugs. This is a case report of a 70-year-old female patient who continued to take edoxaban and suffered bilateral thyroid hematoma a few hours after ultrasound-guided thyroid fine needle aspiration. The patient successfully recovered after undergoing conservative treatment.

## 1. Introduction

With an accuracy range of 79–94.5%, ultrasound-guided fine needle aspiration (FNA) is a simple, safe, and effective method for diagnosing thyroid gland tumors [1,2]. Although recent guidelines and studies have demonstrated that this test is safe and has a low incidence of complications, it carries the risk of damaging the surrounding tissues and causing serious and fatal bleeding. In Republic of Republic of Korea, patient deterioration and the development of thyroid hematoma after ultrasound-guided FNA have been reported, to the point that hospitalized care was necessary [3,4,5]. Most guidelines include no recommendations for post-biopsy care. Only a few suggest that the skin puncture area should be manually pressed for an additional 20–30 min after the examination and re-examined using ultrasound if biopsy-related complications are suspected [1,2]. In addition, a thorough medical history is required to evaluate coagulation function disorders (such as hepatic cirrhosis and chronic kidney disease) and bleeding risk factors, such as the use of antithrombotic agents [6,7].

Herein, the authors present the case of a 70-year-old female patient who underwent continuous edoxaban (30 mg) treatment and developed a wide range of bilateral thyroid hematomas after thyroid ultrasound-guided FNA, ranging from the subgrade to the upper mediastinal region. This study includes the literature on the requirement for a thorough review of the history, discontinuation of anticoagulant treatment and antithrombotic medications that potentially increase the risk of bleeding before the test, information regarding the recommendations required after the test, and conservative treatment options for bilateral thyroid hematomas.

## 2. Case Report

A 70-year-old female patient was initially suspected of having thyroid nodules during a comprehensive medical checkup at the family medicine department and was subsequently referred to the otorhinolaryngology department at our institution. All thyroid function tests conducted during the checkup yielded normal results. Thyroid ultrasonography revealed benign spongy lesions measuring 2 mm in the upper right lobe, as well as 8 and 3.5 mm in the lower part. Completely calcified lesions measuring 4 mm were detected in the lower part, while benign-type spongy lesions measuring < 3 mm and colloidal cysts measuring < 4 mm were observed in the left lobe of the thyroid gland. The patient had previously been administered edoxaban (30 mg) to treat atrial fibrillation. The patient was advised to undergo annual ultrasound follow-up for thyroid nodules; however, the patient preferred a biopsy. After the ultrasound, we explained to the patient that there were no indications for FNA, but the patient expressed a solid willingness to conduct additional tests. Consequently, the patient was scheduled for a biopsy and instructed to discontinue antithrombotic medications for five days. Four months later, before the examination, the patient was instructed to prohibit taking anticoagulants, and the nodule was confirmed using ultrasound. No increase in blood flow was noted in the thyroid parenchyma or nodule, and no blood vessels were observed around the nodule using Doppler ultrasound (Figure 1A,B). The puncture area of the neck was cleansed with a 10% alcohol solution. Holding an ultrasound probe in the left hand and a 10 mL syringe with a 23 G injection needle in the right hand, the thyroid nodule to be biopsied was centered on the ultrasound transverse plane, and the needle was inserted into the ultrasound probe. The needle was gently positioned in the center of the probe, and its position was checked on the ultrasound screen by gently moving it up and down. The patient was instructed to relax her neck, and a transverse needle was inserted into the right nodule (Figure 1C) and used to aspirate twice. No active bleeding was observed on ultrasound. (Figure 1D) Following the biopsy, the biopsy site was gently compressed with manual pressure for minutes and a small bandage was applied. 

The patient was instructed to compress the site of the FNA on her own with an ice pack, and after confirming that there were no signs of swelling or bleeding in the site, she was advised to return home. However, the patient was presented to the emergency room at our institution 3 h later with dyspnea, swelling, and a stuffy throat. The patient was re-evaluated to determine if she had discontinued the anticoagulant. She reported taking edoxaban until the test day without realizing that it was an anticoagulant. The physical examination revealed severe left-right symmetrical expansion and overall tenderness in the foreground (Figure 2A). Physical examination revealed no evidence of chest wall depression or respiratory abnormalities. On laryngoscopy, swelling was observed in the posterior pharyngeal space (Figure 2B), but no vocal cord swelling or paralysis were noted.

Both thyroid lobes exhibited diffuse swelling and heterogeneous contrast enhancement on neck computed tomography. No airway obstruction was observed. The anterior neck, parapharyngeal space, retropharyngeal space, and upper mediastinum exhibited increased soft-tissue density without any contrast enhancement (Figure 3A,B). 

All vital signs and blood test results, including coagulation and arterial blood gas tests, were normal. Ice packs were applied to the neck. A facial mask with an oxygen flow rate of 3 L/min was provided; saturation was monitored using an oxygen saturation monitoring device. Steroid (dexamethasone 5 mg) was injected intravenously to reduce the swelling. Owing to the patient’s older age and continued edoxaban consumption until the day of the examination, the patient was hospitalized to monitor her symptoms on a regular basis using a laryngoscope and oxygen saturation monitor. Anterior swelling had reduced, as observed on physical examination, owing to ice pack application and intravenous steroid administration. Preparation for airway intubation was performed in case of symptom aggravation, and edoxaban treatment was ceased. Retropharyngeal edema based on laryngoscopy on the second day of hospitalization decreased from that on the previous day (Figure 4), and the patient’s dyspnea and dysphagia were also alleviated. 

Furthermore, daily radiographs of the anterior- posterior, and lateral neck were taken, and swelling relief was observed. The patient’s symptoms steadily improved after hospitalization, and edoxaban was resumed on the fourth day after hospitalization owing to the risk of underlying diseases. Most of the hematomas were found to be absorbed on thyroid ultrasound taken on the seventh day after hospitalization; neck swelling improved on physical examination (Figure 5A,B), and the patient’s symptoms completely improved, resulting in their discharge. The patient remained under outpatient follow-up after discharge, with no notable symptoms.

## 3. Discussion

Ultrasound-guided FNA of thyroid nodules is a simple, minimally invasive, and safe test generally performed in ambulatory care [1,8]. The most common complications are local pain or discomfort and mild hematoma [9]. Symptoms of thyroid hematoma include swelling of the neck, dyspnea, hoarseness, dysphagia, and airway deviation. Ultrasound or computed tomography can be performed to diagnose thyroid hematomas, and computed tomography is currently the most reliable diagnostic method for identifying extensive hematomas and damage to tissues around the thyroid gland, esophagus, and airways. It is also important to determine damage to the major blood vessels through angiography to determine the treatment plan. According to various studies, the incidence rate of hematoma ranges between 0% and 6.4%. The most likely factor influencing thyroid bleeding after an examination is venous bleeding within or around the nodule [8,9]. High blood pressure, a sharp decrease in intra-nodular pressure immediately after cystic nodule aspiration, a severe bleeding diathesis, and a history of anticoagulant or antithrombotic medication use are all risk factors for hemorrhagic complications [8,9]. The thyroid gland’s rich blood supply, thyroid nodule’s abnormally thin veins, and intra-nodular arteriovenous shunt that bypasses high-pressure blood through such weakened veins render the patient vulnerable to post-examination bleeding [9]. Using force during the procedure potentially causes central venous pressure to rise, resulting in bleeding. Furthermore, bleeding can occur immediately after body fluid aspiration in complex nodules due to a rapid decrease in nodular pressure caused by body fluid discharge. Additionally, excessive tension during the examination or an increase in blood pressure can increase bleeding. In our case, bleeding inside the thyroid nodule was considered because Doppler ultrasound revealed no blood vessels around the nodule. Patients have previously been instructed to return home if no difference in pain is noted after compressing the insertion site for 15–20 min following the procedure. However, according to one study, changes in pain scores at the aspiration site 30 min after the test are potentially useful as a significant predictor of bleeding [1]. Therefore, on the premise of this case, it is recommended that the patients compress the site of FNA by themselves for about 30 min and check for post-operative pain or changes in swallowing function after returning home. For FNA, a 10cc syringe and a 25 G needle are typically used, and the thinner the needle, the better the tissue penetration and the more effective it is at aspiration. In our case, the possibility of bleeding using a 23 G needle could not be ruled out. In order to prevent bleeding, the patient should refrain from talking, swallowing, or moving during the procedure; moreover, if the direction of the needle is wide (as a fan shape) during FNA, the possibility of bleeding or hematoma increases [4].

Certain reports disagree on whether anticoagulants or antithrombotic medication should be avoided before ultrasound-guided fine needle aspiration. Some studies have revealed that taking these drugs has no effect on the post-examination hematoma incidence rate; therefore, discontinuing the drug prior to testing is not necessary [10,11,12]. In contrast, other studies recommend discontinuing warfarin, aspirin, or clopidogrel the to five days before FNA [2]. Some researchers have also advocated for the continued use of a recently developed oral anticoagulant, called new oral anticoagulant [11,12]. Edoxaban (30 mg) was being taken by the patient as mentioned above. Unlike in previous studies, severe bleeding occurred after continuous drug use. As such, considering the patient’s underlying diseases or medical history, it is recommended not to take anticoagulants or antithrombotic drugs, although this approach might not be suitable for all patients. One week before the test, we instruct patients to stop taking the medications by phone and confirm it again on the day of the test. Furthermore, if the test is to be conducted in cases where the anticoagulant and antithrombotic drugs cannot be stopped owing to the patient’s underlying disease, the risk of bleeding must be confirmed using a blood coagulation test. Moreover, even in patients with a high risk of bleeding, such as those with chronic liver disease and renal failure, blood coagulation tests are necessary prior to the test. 

With regard to thyroid hematoma, if the bleeding is accompanied by dyspnea or unstable vital signs, the cause of the bleeding should be determined surgically. Airway intubation should be preferentially considered in severe cases where upper airway obstruction is suspected [13]. In two studies, thyroidectomy and hematoma removal were conducted, whereas in another, treatment was performed via intravascular intervention [4,5,14]. In addition, there was one case wherein airway intubation was performed as a conservative treatment for thyroid hematoma caused by neck trauma [13]. However, surgical treatment has the disadvantage of increasing the risk of recurrent laryngeal nerve damage, diaphragm nerve damage, and wound infection. The risk of mortality due to low blood volume from excessive blood loss during surgery exists in patients who exhibit a high tendency to bleed while taking anticoagulants or antithrombotic drugs [14]. The hematoma’s pressure can naturally stop bleeding, and the hematoma is also considerably likely to be in spontaneous remission; therefore, it can be treated conservatively if the airway is well maintained [13]. There are a few studies that have focused on the determinants for choosing between surgical and conservative treatment, and otolaryngologists can assist with decision-making for airway management. The primary requirement for conservative treatment is to ensure that epiglottis and vocal cords are visible on laryngoscopy. Furthermore, a comprehensive evaluation is necessary to determine the degree of dyspnea, range of the hematoma expansion, and to identify the patient’s age, medical history, and anticoagulant use. If the patient is in a generally good condition and has stable vital signs, even extensive hematomas could be treated effectively by conservative treatment [15]. However, if delayed onset of the airway compression occurs, intensive monitoring may be warranted [16]. In this case, surgical treatment was initially considered after the patient complained of symptoms, such as dyspnea and change in voice during the visit. However, conservative treatment was selected based on the risk of surgical bleeding while taking anticoagulants, and the patient had a strong desire for conservative treatment, follow-up observation in the emergency room and during endoscopy, and symptom improvement. Following computed tomography, the retropharyngeal space was monitored every 2 h using a laryngoscope until the day after hospitalization, and the patient was asked if the symptoms had improved. Furthermore, daily radiography of the anterior-posterior and lateral neck was performed to check for anterior and retropharyngeal soft-tissue inflammation and edema subsidence. Steroid injections were also administered to reduce laryngeal edema. As described above, FNA is an effective method for the diagnosis of thyroid nodules, but it is necessary to inform the patient prior to the test of rare fatal complications, such as hematoma or intracapsular hemorrhage. It is crucial that a sufficient explanation be made. This is the first case of successful conservative treatment without surgical treatment for thyroid hematoma with dyspnea after ultrasound-guided FNA to be reported in Korea. It is suggested that, in clinically similar cases, conservative treatment methods be considered in addition to surgical treatment.

## Figures and Tables

**Figure 1 medicina-59-00690-f001:**
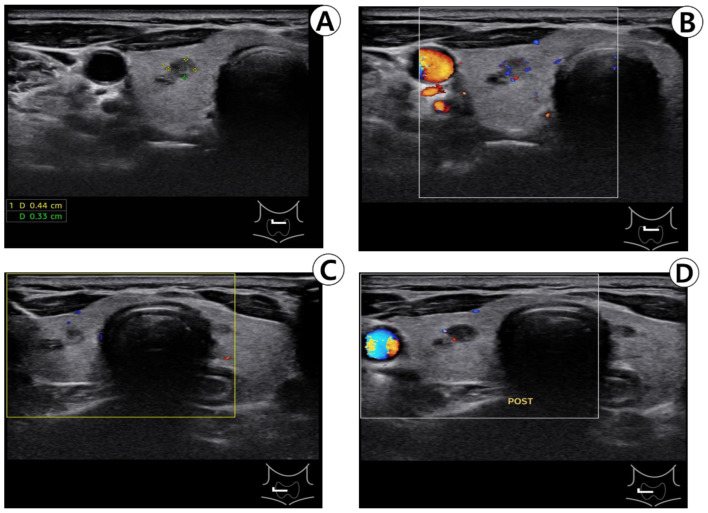
Ultrasonography (US) of the thyroid before (**A**,**B**) and after (**C**,**D**) fine-needle aspiration (FNA). (**A**) US before FNA, revealing a solid and hypoechoic nodule in the right lobe. (**B**) Color-Doppler US revealing a solid, hypoechoic nodule without increase in vascularity that could cause potential bleeding during FNA. (**C**,**D**) On immediate US after biopsy, there was no evidence of post-biopsy hemorrhage.

**Figure 2 medicina-59-00690-f002:**
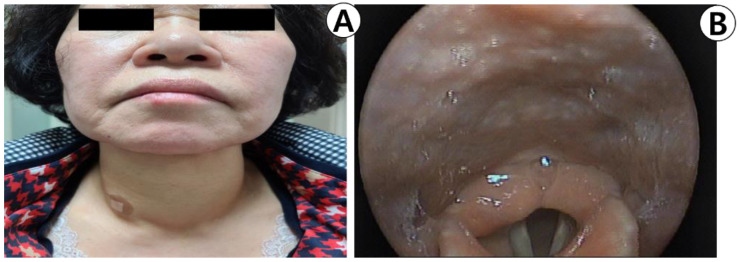
(**A**) Photograph of the patient in the emergency room. Symmetrical diffuse swelling of the anterior neck is observed. (**B**) Laryngoscopic view of retropharyngeal wall swelling on admission day.

**Figure 3 medicina-59-00690-f003:**
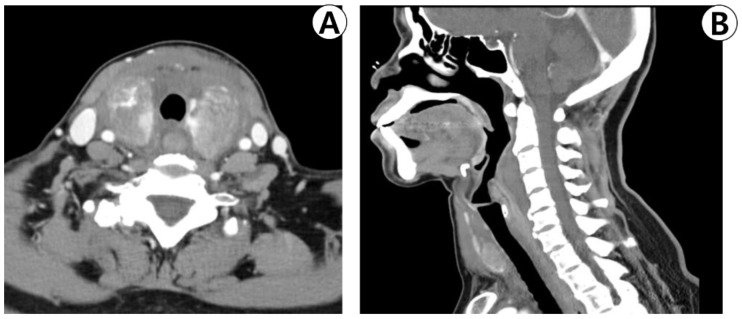
Enhanced neck computed tomography. (**A**) The axial view shows diffuse swelling and heterogenous enhancement of both thyroid lobes. (**B**) The sagittal view shows diffuse non-enhancing soft-tissue density lesions of the retropharyngeal space and superior mediastinum.

**Figure 4 medicina-59-00690-f004:**
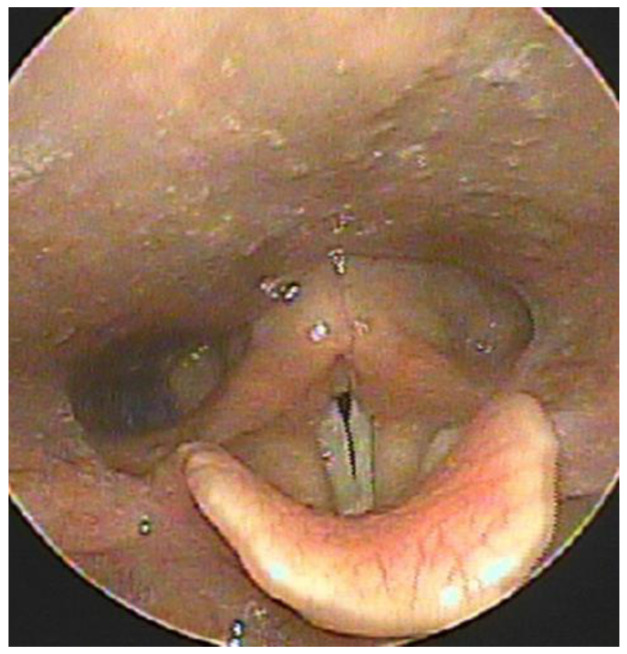
Laryngoscopic view of improved retropharyngeal swelling on the second day of admission.

**Figure 5 medicina-59-00690-f005:**
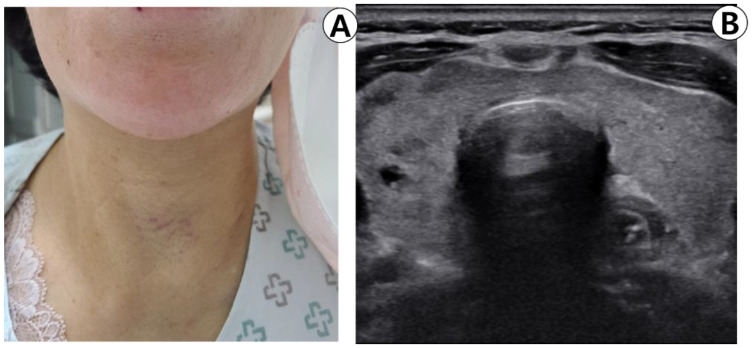
(**A**) Photograph of the patient on discharge day showing improved swelling of the anterior neck. (**B**) Sonographic view of improved thyroid hematoma on discharge day.

## Data Availability

Not applicable.

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
