# Peer review of "A Case of Diffuse Thyroid Hematoma after Ultrasound-Guided Fine Needle Aspiration"

_medicina, 2023, doi:10.3390/medicina59040690_

Round 1
Reviewer 1 Report
This article presents a case of diffuse thyroid hematoma following thyroid FNA, leading the authors to conclude that anticoagulation should be held prior to FNA, and thorough history regarding appropriate discontinuation of medications is completed prior to the procedure. They also discuss the role for conservative treatment in management of post procedure thyroid hematomas.
The recommendation to discontinue anticoagulants is not consistent with current guidelines, and highlights some of the discrepencies in the literature regarding this topic. However, there are several other factors that may have contributed to the patient's hematoma in this case that were not adequately addressed.
Primarily is there was no indication for FNA in this patient. As the authros noted, the nodules were all small and very low risk. They would not meet FNA criteria using any of the current guidelines. Thus, avoiding FNA in a patient with no indication who is on anticoagulants would be important to address, and this was not mentioned in their article.
Also, the size of the needle used for the FNA (23G in this case) was larger than necessary, and while the authors did comment that a 25G needle is standard, they did not address the possibility that the larger needle may have contribtued to the complications.
Thus it is difficult to make the jump to recommending all patients hold anticoagulation prior to FNA (lines 186-187) without addressing these other factors.
The part of the case discussing conservative management of thyroid hematoma is much stronger, and authors may want to consider focusing on this element of the case for the discussion, as this may be more useful to a broad community.
The article would also benefit from editing to make the article more concise, and grammatically consistent throughout.
Author Response
REVIEWER COMMENTS:
# Reviewer 1
|
Point 1 |
The recommendation to discontinue anticoagulants is not consistent with current guidelines, and highlights some of the discrepencies in the literature regarding this topic. However, there are several other factors that may have contributed to the patient's hematoma in this case that were not adequately addressed. |
|
Answer |
We thank the reviewer for bringing up this point. While the authors agree that the current guidelines recommend that the test is carried out while maintaining anticoagulants, the advantage of this report is that even if sono-guided FNA is done without stopping taking edoxaban, thyroid hematoma has been found. |
|
Point 2 |
Primarily is there was no indication for FNA in this patient. As the authros noted, the nodules were all small and very low risk. They would not meet FNA criteria using any of the current guidelines. Thus, avoiding FNA in a patient with no indication who is on anticoagulants would be important to address, and this was not mentioned in their article. |
|
|
Answer |
Thank you for the comment. We agree with your point of view. After the ultrasound, we explained to the patient that there were no indications for FNA, but the patient expressed a solid willingness to conduct additional test. The text has now been added to remove any ambiguity. |
|
|
Before |
After revision |
|
|
The patient was advised to undergo annual ultrasound follow-up for thyroid nodules; however, the patient preferred a biopsy. Consequently, the patient was scheduled for a biopsy and instructed to discontinue antithrombotic medications for 5 days. |
The patient was advised to undergo annual ultrasound follow-up for thyroid nodules; however, the patient preferred a biopsy. After the ultrasound, we explained to the patient that there were no indications for FNA, but the patient expressed a solid willingness to conduct additional test. Consequently, the patient was scheduled for a biopsy and instructed to discontinue antithrombotic medications for 5 days. |
|
|
Point 3 |
Also, the size of the needle used for the FNA (23G in this case) was larger than necessary, and while the authors did comment that a 25G needle is standard, they did not address the possibility that the larger needle may have contribtued to the complications. |
|
|
Answer |
We agree with the reviewer’s advice and therefore The authors added that the possibility of bleeding using a 23g needle cannot be ruled out. The discussion section been updated to include this constraint. |
|
|
Before |
After revision |
|
|
For fine needle aspiration, a 10cc syringe and a 25 gauge needle are mainly used, and the thinner the needle, the better the tissue penetration and the more effective it is to aspirate cells. In order to prevent bleeding, the patient should refrain from talking, swallowing, or moving during the procedure, and even if the direction of the fine needle is wide in a fan shape during fine needle aspiration, the possibility of bleeding or hematoma increases. |
For fine needle aspiration, a 10cc syringe and a 25 gauge needle are mainly used, and the thinner the needle, the better the tissue penetration and the more effective it is to aspirate cells. In our case, the possibility of bleeding using a 23g needle cannot be ruled out. In order to prevent bleeding, the patient should refrain from talking, swallowing, or moving during the procedure, and even if the direction of the fine needle is wide in a fan shape during fine needle aspiration, the possibility of bleeding or hematoma increases. |
|
|
Point 4 |
Thus it is difficult to make the jump to recommending all patients hold anticoagulation prior to FNA (lines 186-187) without addressing these other factors. |
|
|
Answer |
We thank the reviewer for the remarks on our report. With respect to the reviewer’s request for the section, The text has now been modified to remove any ambiguity. “Since then, considering the patient's underlying diseases or past history, it has been recommended not to take anticoagulants or antithrombotic drugs, although the same treatment might not be suitable for all patients” |
|
|
Before |
After revision |
|
|
Since then, patients have been advised to discontinue anticoagulants or antithrombotic drugs prior to the test, considering the half-life of each drug and underlying disease. |
Since then, considering the patient's underlying diseases or past history, it has been recommended not to take anticoagulants or antithrombotic drugs, although the same treatment might not be suitable for all patients. |
|
|
Point 5 |
The part of the case discussing conservative management of thyroid hematoma is much stronger, and authors may want to consider focusing on this element of the case for the discussion, as this may be more useful to a broad community. |
|
|
Answer |
Thank you for your review. Your efforts are most appreciated. The discussion section been updated to include this element with 2 references. “There are few studies which have focused on the determinants for choosing between surgical and conservative treatment. Otolaryngologists can help on decision-making for airway management. The primary requirement for conservative treatment is to ensure that epiglottis and vocal cords are visible on laryngoscopy. Furthermore, a comprehensive evaluation is necessary such as the degree of dyspnea, range of the hematoma expansion, and patient’s age, medical history, anticoagulants administration.[15] If the patient is in a generally good condition and has stable vital sign, even extensive hematomas could be cured by conservative treatment. However delayed onset of the airway compression up to one day or even longer would occur, so intensive monitoring for more than one day may be warranted.[16]” |
|
|
Before |
After revision |
|
|
The hematoma’s pressure can naturally stop bleeding, and the hematoma is also consid-erably likely to be in spontaneous remission; therefore, it can be treated conservatively if the airway is well maintained [13]. In this case, surgical treatment was initially considered after the patient complained of symptoms such as dyspnea and change in voice during the visit. |
The hematoma’s pressure can naturally stop bleeding, and the hematoma is also consid-erably likely to be in spontaneous remission; therefore, it can be treated conservatively if the airway is well maintained [13]. There are few studies which have focused on the determinants for choosing between surgical and conservative treatment. Otolaryngologists can help on decision-making for airway management. The primary requirement for conservative treatment is to ensure that epiglottis and vocal cords are visible on laryngoscopy. Furthermore, a comprehensive evaluation is necessary such as the degree of dyspnea, range of the hematoma expansion, and patient’s age, medical history, anticoagulants administration. [15] If the patient is in a generally good condition and has stable vital sign, even extensive hematomas could be cured by conservative treatment. However delayed onset of the airway compression up to one day or even longer would occur, so intensive monitoring for more than one day may be warranted.[16] In this case, surgical treatment was initially considered after the patient complained of symptoms such as dyspnea and change in voice during the visit. |
|
|
Point 6 |
The article would also benefit from editing to make the article more concise, and grammatically consistent throughout. |
|
Answer |
We apologize for the confusion and thank you for pointing out this problem.
|

Reviewer 2 Report
This article presents an interesting case which may be useful for similar situations considering that the preformed exam should not induce severe complications. The case is well presented although a few information needs to be clarified.
Below I present some questions and comments, as well as suggestions for minor corrections, that I believe may be useful to improve the quality of the article presented.
Line 43: the department you refer should be “Otorhinolaryngology”, right? As in your affiliation?
Line 56: review letter size
Paragraph (line 56-67): review verbal tense (some sentences are written in the present tense. Should they be presented in the past tense?)
Lines 60-61: “Lightly position the probe in the center of the probe without touching it…” position the probe in the center of the probe is correct? Or do you mean position the needle?
Line 85-86: “The patient was instructed to stop bleeding for 15 min.” What do you mean by this? How can a patient be instructed to stop bleeding? Then you refer that the patient had no special symptoms. What do you mean by “special”; did she present other symptoms (not considered special)?
Lines 89-90: “claiming that they could not discontinue edoxaban until the morning of the examination because it was not recognized as an anticoagulant.” – after reading line 118 (“and continued edoxaban treatment until the day of the examination) it is not clear to me if the patient did stop taking the medication before the exam, as required, or if she continued the treatment, because you also say (lines 53-54) that “Four months later, after verifying if the antithrombotic drug had been discontinued in the outpatient clinic,…” Please clarify this question throughout the text.
Line 171-172: for potential readers less familiar with these terms, what does it mean “…stopping bleeding for 30 min… after returning home…”? What should the patient do?
It seems to me that there are several statements in the discussion that are presented based on the authors’ experience, and not in literature references. For example, lines 173-178. Are there any references that may be cited?
Line 185: “… recently developed oral anticoagulant new oral anticoagulant (NOAC)…” Is this correct?
Line 189-190: “Whether they have ceased taking them on the day of the procedure is routinely confirmed.” Can you describe how do you confirm this?
Line 187-195: the prevention measures described in this paragraph are applied in the institution where the authors are affiliated or are standard measures in other institutions too?
Author Response
REVIEWER COMMENTS:
# Reviewer 2
|
Point 1 |
Line 43: the department you refer should be “Otorhinolaryngology”, right? As in your affiliation? |
|
|
Answer |
Thank you for the comment and we fully agree with this point. The text has now been modified to remove any ambiguity. Amended to now read: A 70-year-old female patient was initially suspected of having thyroid nodules during a comprehensive medical checkup at the family medicine department and was subsequently referred to the otorhinolaryngology department at our institution. |
|
|
Before |
After revision |
|
|
A 70-year-old female patient was suspected of having thyroid nodules during a comprehensive medical checkup and was subsequently referred to the otolaryngology department at our institution |
A 70-year-old female patient was initially suspected of having thyroid nodules during a comprehensive medical checkup at the family medicine department and was subsequently referred to the otorhinolaryngology department at our institution |
|
|
Point 2 |
Line 56: review letter size |
|
Answer |
Thank you for the comment. The letter size has been modified to suit the formatting guide of the article |
|
Point 3 |
Paragraph (line 56-67): review verbal tense (some sentences are written in the present tense. Should they be presented in the past tense?) |
|
|
Answer |
Thank you for pointing the error. This sentence has now been amended for better clarity |
|
|
Before |
After revision |
|
|
Holding an ultrasound probe in the left hand and a 10 mL syringe with a 23 G injection needle in the right hand, reassuring the patient, adjust the thyroid nodule to be biopsied in the center of the ultrasound transverse plane, and insert the needle of the syringe into the ultrasound probe. |
Holding an ultrasound probe in the left hand and a 10 mL syringe with a 23 G injection needle in the right hand, reassuring the patient, adjusted the thyroid nodule to be biopsied in the center of the ultrasound transverse plane, and inserted the needle of the syringe into the ultrasound probe. |
|
|
Point 4 |
Lines 60-61: “Lightly position the probe in the center of the probe without touching it…” position the probe in the center of the probe is correct? Or do you mean position the needle? |
|
|
Answer |
We apologize for the confusion and thank you for pointing out this problem. This has now been amended to ‘Lightly position the needle in the center of the probe’. |
|
|
Before |
After revision |
|
|
Lightly position the probe in the center of the probe |
Lightly position the needle in the center of the probe |
|
|
Point 5 |
Line 85-86: “The patient was instructed to stop bleeding for 15 min.” What do you mean by this? How can a patient be instructed to stop bleeding? Then you refer that the patient had no special symptoms. What do you mean by “special”; did she present other symptoms (not considered special)? |
|
|
Answer |
Thank you for your feedback.You have raised an important point here. This has now been corrected as suggested. The patient was instructed to compress the site of fine needle aspiration on her own with an ice pack, and after confirming that there were no signs of swelling or bleeding in the site, she was advised to return home. |
|
|
Before |
After revision |
|
|
The patient was instructed to stop bleeding for 15 min. After hemostasis, the patient was discharged with no special symptoms. |
The patient was instructed to compress the site of fine needle aspiration on her own with an ice pack, and after confirming that there were no signs of swelling or bleeding in the site, she was advised to return home. |
|
|
Point 6 |
Lines 89-90: “claiming that they could not discontinue edoxaban until the morning of the examination because it was not recognized as an anticoagulant.” – after reading line 118 (“and continued edoxaban treatment until the day of the examination) it is not clear to me if the patient did stop taking the medication before the exam, as required, or if she continued the treatment, because you also say (lines 53-54) that “Four months later, after verifying if the antithrombotic drug had been discontinued in the outpatient clinic,…” Please clarify this question throughout the text. |
|
|
Answer |
We think this is an excellent comment and have incorporated your suggestion.. The text has now been modified to remove any ambiguity. This sentence has now been amended to be more exact: lines 53-54 Before the examination, the patient was required to prohibit taking anticoagulants. Lines 89-90 After going to the emergency room, the patient was rechecked to determine whether or not she had discontinued the anticoagulant. She reported taking edoxaban until the test day without knowing it was an anticoagulant. |
|
|
Before |
After revision |
|
|
lines 53-54 Four months later, after verifying if the antithrombotic drug had been discontinued in the outpatient clinic, the nodule was confirmed using ultrasound
Line 89-90 The patient complained of dyspnea and wheezing in the emergency room, claiming that they could not discontinue edoxaban until the morning of the examination because it was not recognized as an anticoagulant. |
lines 53-54 Four months later, before the examination, the patient was required to prohibit taking anti-coagulants and the nodule was confirmed using ultrasound
Line 89-90 The patient complained of dyspnea and wheezing in the emergency room After going to the emergency room, the patient was rechecked to determine whether or not she had discontinued the anticoagulant. She reported taking edoxaban until the test day without knowing it was an anticoagulant. |
|
|
Point 7 |
Line 171-172: for potential readers less familiar with these terms, what does it mean “…stopping bleeding for 30 min… after returning home…”? What should the patient do? |
|
|
Answer |
We thank the reviewer for the remarks on this point. This sentence has now been amended to be more exact as below: It is recommended that the patients compress the site of fine needle aspiration on his or her own for about 30 minutes and check for post-operative pain or changes in swallowing levels after returning home. |
|
|
Before |
After revision |
|
|
Therefore, on the premise of this case, stopping bleeding for 30 min, checking for changes in pain, and checking for changes in swelling at the insertion site after returning home are recommended. |
Therefore, on the premise of this case, It is recommended that the patients compress the site of fine needle aspiration on his or her own for about 30 minutes and check for post-operative pain or changes in swallowing levels after returning home. |
|
|
Point 8 |
It seems to me that there are several statements in the discussion that are presented based on the authors’ experience, and not in literature references. For example, lines 173-178. Are there any references that may be cited? |
|
|
Answer |
The authors completely agree with reviewer’s comment. We acknowledge that the discussion of related work was incomplete. References to earlier work by author Byun, J.-Y (2007) have now been included. |
|
|
Before |
After revision |
|
|
In order to prevent bleeding, the patient should refrain from talking, swallowing, or moving during the procedure, and even if the direction of the fine needle is wide in a fan shape during fine needle aspiration, the possibility of bleeding or hematoma increases. |
In order to prevent bleeding, the patient should refrain from talking, swallowing, or moving during the procedure, and even if the direction of the fine needle is wide in a fan shape during fine needle aspiration, the possibility of bleeding or hematoma increases.[4] |
|
|
Point 9 |
Line 185: “… recently developed oral anticoagulant new oral anticoagulant (NOAC)…” Is this correct? |
|
|
Answer |
Thank you for pointing this out. The reviewer is correct. This should have read ‘recently developed oral anticoagulant, called new oral anticoagulant (NOAC) ’ and has now been amended. |
|
|
Before |
After revision |
|
|
Some researchers have also advocated for the continued use of the recently developed oral anticoagulant new oral anticoagulant (NOAC) |
Some researchers have also advocated for the continued use of the recently developed oral anticoagulant, called new oral anticoagulant (NOAC). |
|
|
Point 10 |
Line 189-190: “Whether they have ceased taking them on the day of the procedure is routinely confirmed.” Can you describe how do you confirm this? |
|
|
Answer |
We sincerely appreciate the reviewer’s comments. The text has now been amended to provide better clarification: A week before test, we instruct patients to stop taking medicine by phone and confirm it again on the day of the procedure. |
|
|
Before |
After revision |
|
|
Whether they have ceased taking them on the day of the procedure is routinely confirmed. |
A week before test, we instruct patients to stop taking medicine by phone and confirm it again on the day of the procedure. |
|
|
Point 11 |
Line 187-195: the prevention measures described in this paragraph are applied in the institution where the authors are affiliated or are standard measures in other institutions too? |
|
Answer |
Thank you for the suggestion. For clarification, the author’s affiliation uses those specifically listed prevention measures in the paragraph. |

Round 2
Reviewer 1 Report
The refocus of the case report on conservative management of thyroid hematoma is very good, and I appreciate their response to my initial comments. The discussion is a bit long, and could be edited to be more concise and direct, but overall, no significant concerns or revisions needed.